# MODEL COMPRESSION WITH GENERATIVE ADVERSARIAL NETWORKS

## ABSTRACT

More accurate machine learning models often demand more computation and memory at test time, making them difficult to deploy on CPU- or memory-constrained devices. *Model compression* (also known as *distillation*) alleviates this burden by training a less expensive student model to mimic the expensive teacher model while maintaining most of the original accuracy. However, when fresh data is unavailable for the compression task, the teacher's training data is typically reused, leading to suboptimal compression. In this work, we propose to augment the compression dataset with synthetic data from a generative adversarial network (GAN) designed to approximate the training data distribution. Our *GAN-assisted model compression* (GAN-MC) significantly improves student accuracy for expensive models such as deep neural networks and large random forests on both image and tabular datasets. Building on these results, we propose a comprehensive metric—the *Compression Score*—to evaluate the quality of synthetic datasets based on their induced model compression performance. The Compression Score captures both data diversity and discriminability, and we illustrate its benefits over the popular Inception Score in the context of image classification.

## 1 INTRODUCTION

Modern machine learning models have achieved remarkable levels of accuracy, but their complexity can make them slow to query, expensive to store, and difficult to deploy for real-world use. Ideally, we would like to replace such cumbersome models with simpler models that perform equally well. One way to address this problem is to perform *model compression* (also known as *distillation*), which consists of training a student model to mimic the outputs of a teacher model (Bucila et al., 2006; Hinton et al., 2015). For example, expensive ensemble and deep neural network (DNN) teachers have been used to train inexpensive decision tree (Craven & Shavlik, 1996; Frosst & Hinton, 2017) and shallow neural network (Bucila et al., 2006; Ba & Caruana, 2014; Hinton et al., 2015; Urban et al., 2017) students.

An important degree of freedom in the model compression problem is the *compression set*[1] used to train the student. Ideally, fresh (unlabeled) data from the training distribution would fuel this task, but often no fresh data remains after the teacher is trained (Bucila et al., 2006; Ba & Caruana, 2014). In this case, one branch of the literature, dating back to the pioneering work of Bucila et al. (2006), recommends generating synthetic data for compression and proposes tailored generation schemes for tabular (Bucila et al., 2006) and image (Urban et al., 2017) data. A second branch, rooted in the distillation community (Hinton et al., 2015; Frosst & Hinton, 2017), simply uses the same data to train teacher and student (see also Ba & Caruana, 2014). Here, we show that the latter convention leads to suboptimal compression performance and propose a synthetic data generation strategy that is suitable for image and tabular data alike.

Specifically, when fresh data is unavailable for model compression, we propose to augment the compression set with synthetic data produced by generative adversarial networks (GANs) (Goodfellow et al., 2014). GANs attempt to generate new datapoints from the distribution underlying a given dataset and have achieved impressive performance for a variety of data types including images

---

[1]To avoid ambiguity, we will refer to the dataset used for compression as "compression set" and reserve the name "training set" for the data used to train the teacher.

(Goodfellow et al., 2014), text (Yu et al., 2017), and electronic health records (Choi et al., 2017). Here, we develop *GAN-assisted model compression* (GAN-MC) to improve the compression of expensive machine learning classifiers and demonstrate its effectiveness for both image and tabular data.

We further propose to use GAN-MC to evaluate the quality of synthetic datasets and their generators. In essence, we declare a synthetic dataset to be of higher quality if a compressed model trained on that data achieves higher test accuracy. Synthetic data evaluation is a notoriously difficult problem marked by the lack of universally agreed-upon quality measures (Theis et al., 2015). Some standard quality measures, like *multiscale structural similarity* (Wang et al., 2003), quantify the diversity of a synthetic dataset but do not capture *discriminability*, the ability of datapoints to be correctly associated with their labels with high confidence. Others, like the popular *Inception Score* (Salimans et al., 2016), quantify discriminability based on the predicted label distribution of a trained neural network. However, these scores do not account for within-class diversity and are easily misled by adversarial datapoints that elicit high confidence predictions but do not resemble real data.

To address these shortcomings, we develop a *Compression Score* that quantifies the true test accuracy of compressed models trained using synthetic data; this offers a robust, goal-driven metric for synthetic data quality that accounts for both diversity and discriminability. In summary, we make the following principal contributions in this paper:

1. We propose GAN-assisted model compression (GAN-MC), a simple approach to improving teacher-student compression by augmenting the compression set with GAN data.

2. On CIFAR-10 image classification, we show GAN-MC consistently improves student test accuracy for a variety of deep neural network teacher-student pairings and two popular compression objectives.

3. For random forest teachers, we demonstrate 25 to 336-fold reductions in execution and storage costs with less than $1.2\%$ loss in test performance across a suite of real-world tabular datasets.

4. We introduce a new Compression Score for evaluating the quality of GAN-generated datasets and illustrate its advantages over the popular Inception Score on CIFAR-10.

## 2    MODEL COMPRESSION WITH GANS

In this section, we review standard approaches to model compression for DNNs and describe our proposals for compressing random forests and improving model compression with GAN data.

### 2.1    DEEP NEURAL NETWORK COMPRESSION

In the standard teacher-student approach to compressing a neural network classifier, a relatively inexpensive prediction rule, like a shallow neural network, is trained to predict the unnormalized log probability values—the *logits* $z$—assigned to each class by a previously trained deep network classifier. The inexpensive model is termed the *student*, and the expensive deep network is termed the *teacher*. Given a compression set of $n$ feature vectors paired with teacher logit vectors, $\{(x^{(1)}, z^{(1)}), ..., (x^{(n)}, z^{(n)})\}$, Ba & Caruana (2014) proposed framing the compression task as a multitask regression problem with $L^2$ loss,

$$L(\theta) = ||g(x; \theta) - z||_2^2. \tag{1}$$

Here, $\theta$ represents any student model parameters to be learned (e.g., the student network weights), and $g(x; \theta)$ is the vector of logits predicted by the student model for the input feature vector $x$.

Hinton et al. (2015) introduced an alternative compression objective function, indexed by a temperature parameter $T > 0$. Specifically, the student is trained to mimic the annealed teacher class probabilities,

$$q_j(z/T) = \frac{\exp(z_j/T)}{\sum_k \exp(z_k/T)},$$

for each class $j$ by solving a multitask regression problem with cross-entropy loss,

$$L_T(\theta) = -\textstyle\sum_j q_j(z/T) \log(q_j(g(x; \theta)/T)).$$

Hinton et al. (2015) showed that, under a zero-mean logit assumption, cross-entropy regression recovers $L^2$ logit matching as $T \to \infty$; however, the two approaches can differ for small $T$. In Sec. 3, we will experiment with both of these popular compression approaches.

## 2.2 RANDOM FOREST COMPRESSION

Random forests (Breiman, 2001) construct highly accurate prediction rules by averaging the predictions of a diverse and often large collection of learned decision trees. Effectively mimicking a large random forest with a single decision tree or a small forest has the potential to reduce prediction computation and storage costs by multiple orders of magnitude (Bucila et al., 2006; Joly et al., 2012; Begon et al., 2017; Painsky & Rosset, 2016; 2018). Focusing on the common setting of binary classification with labels in $\{0, 1\}$, we propose to train a student regression random forest to predict a teacher forest's outputted probability $p$ of a datapoint $x$ having the label 1.

## 2.3 GAN-ASSISTED MODEL COMPRESSION (GAN-MC)

In a typical compression setting, as much data as possible has been dedicated to training the highly accurate teacher model, leaving little fresh data for training the student model. While one branch of the model compression literature recommends generating synthetic data with customized augmentation algorithms for tabular Bucila et al. (2006) and image Urban et al. (2017) data, the more common solution in the distillation literature is to simply reuse the teacher training set as the compression set (Hinton et al., 2015; Frosst & Hinton, 2017). However, we will see in Secs. 3 and 4 that compressing with training data alone leads to suboptimal student performance. To boost student performance and compression efficiency, we propose a simple solution applicable to tabular and image data alike: augment the compression set with synthetic feature vectors generated by a high-quality GAN. These synthetic feature vectors are then labeled with the true outputted teacher class probabilities or logits, as described in Secs. 2.1 and 2.2. We call this approach *GAN-assisted model compression* (GAN-MC).

**Intuition.** Ideally, the student would be trained to mimic the predictions of the teacher on fresh feature vectors drawn from the true data distribution. However, synthetic data with a similar distribution can provide an effective surrogate for training an accurate student. The generator of a GAN for instance can produce an auxiliary stream of fake feature vectors by transforming independent noise vectors drawn from a simple distribution. The distributions of the synthetic and real data are encouraged to align via an adversarial game between a generator and a discriminator.

To generate high-quality GAN feature vectors which capture the salient features of each class, we use the auxiliary classifier GAN (AC-GAN) of Odena et al. (2017). The AC-GAN generator $G$ produces a synthetic feature vector $X_{fake} = G(W, C)$ given a random noise vector $W$ and an independent target class label $C$ drawn from the real data class distribution. For any given feature vector $x$, the AC-GAN discriminator $D$ predicts both the probability of each class label $P(C \mid x)$ and the probability of the data source being real or fake, $P(S \mid x)$ for $S \in \{real, fake\}$. Given a training dataset $\mathcal{D}_{real}$ of labeled feature vectors, two components contribute to the AC-GAN training objective:

$$L_{source} = \frac{1}{|\mathcal{D}_{real}|}\sum_{(x,c)\in\mathcal{D}_{real}} \log P(S = real \mid x) + \mathbb{E}_{W,C\sim p_c}[\log P(S = fake \mid G(W,C))] \text{ and}$$

$$L_{class} = \frac{1}{|\mathcal{D}_{real}|}\sum_{(x,c)\in\mathcal{D}_{real}} \log P(C = c \mid x) + \mathbb{E}_{W,C\sim p_c}[\log P(C \mid G(W,C))], \quad (2a)$$

representing the expected conditional log-likelihood of the correct source and the correct class of a feature vector, respectively. In the adversarial game, the generator $G$ is trained to maximize $L_{class} - L_{source}$, and the discriminator $D$ is trained to maximize $L_{class} + L_{source}$.

It should be noted that there is an important distinction between training a student to mimic a teacher with GAN data and training a student to solve the original supervised learning problem with GAN data. The goal of the original supervised learning task is to approximate the ideal mapping $f^*$ between inputs x and outputs y. This ideal $f^*$ is a functional of the true but unknown distribution underlying our data, and our information concerning $f^*$ is limited by the real data we have collected. The goal in model compression is to approximate the teacher prediction function $g$ which maps from inputs to predictions $z$. Because the teacher is a function of the training data alone, $g$ itself is a functional of the training data alone and is otherwise independent of the unknown distribution

that generated that data. In addition, because we have access to the teacher, we have the freedom to query the function $g$ at any point, and hence our information concerning $g$ is limited only by number of queries we can afford. In particular, when we generate a new query point $x$, we can observe the actual target value of interest, the teacher's prediction $g(x)$; this is not true however for the supervised learning task, where no new labels can be observed. We believe these properties make the model compression task a much more tractable one and one that is ideal for data augmentation with generative models. For further discussion on the distinctions between model compression and the original supervised learning task, we refer the reader to (Ba & Caruana, 2014).

## 3 Deep Neural Network GAN-MC

We now investigate how GAN-MC performs when used to compress convolutional deep neural network (CNN) classifiers trained on the CIFAR-10 dataset of Krizhevsky & Hinton (2009). CIFAR-10 consists of $32 \times 32$ RGB images from 10 classes, divided into 50,000 training and 10,000 test images. The test images are randomly divided into a validation set with size 5000 and a test set with size 5000. The AC-GAN is implemented in Keras (Chollet et al., 2015) and trained for 1000 epochs (Tuya, 2017). The discriminator $D$ is a CNN with 6 convolution layers and Leaky ReLU nonlinearity. The generator $G$ consists of 3 'deconvolution' layers which transform the class $c$ and noise vector $w \in \mathbb{R}^{110}$ into a $32 \times 32$ image with 3 color channels. We use the Adam optimizer with learning rate 0.0002 and momentum term $\beta_1 = 0.5$, as suggested by Radford et al. (2015).

We experiment with both of the compression objectives introduced in Sec. 2.1 using 200 compression training epochs. For $L^2$ logit matching, the teacher and the student are NIN (Lin et al., 2014) and LeNet (LeCun et al., 1998) models. The uncompressed networks are pre-trained by Caffe (Chan, 2016; Jia et al., 2014). Similar to Chan (2016), for compression training, we use the Adam optimizer in Tensorflow (Abadi et al., 2015) with learning rate $10^{-4}$ and the $L^2$ loss in Eq. 1.

For cross-entropy regression, we examine three additional networks: WideResNet-28-10 (Zagoruyko & Komodakis, 2016), ResNet-18 (He et al., 2016) and a 5-layer CNN with 3 convolution layers. Network training both with and without compression is carried out in Pytorch (Li, 2018; Paszke et al., 2017). For compression, we use the student objective $L(\theta) = \alpha L_T(\theta) + (1-\alpha)L_0(\theta)$, where $0 < \alpha \leq 1$ and $L_0(\theta) = -\sum_j \mathbf{1}\{j = c\} \log(q_j(g(x;\theta)))$ is the cross-entropy classification loss for a datapoint $x$ with class label $c$. For each teacher-student pair, we set $T$, $\alpha$, and all optimizer hyperparameters to the default values recommended in (Li, 2018). For the teacher-student pairs 1, 2, and 3 in Table 1, this yields the respective $T$ values 5, 20, and 6 and the respective $\alpha$ values 0.9, 0.9, and 0.95. The Adam optimizer with learning rate $10^{-3}$ is used for the first two teacher-student pairs, and the stochastic gradient descent optimizer with learning rate decayed from 0.1 is used for the third pair.

We compare the standard approach of compression using only the teacher's training dataset to two versions of GAN-MC: compression using only GAN data and compression using a mixture of training and GAN data. The GAN data is produced in real time during the stochastic optimization training. The mixture of training and GAN data is realized by generating GAN data with probability $p_{fake}$ and by sampling from the training set with probability $1 - p_{fake}$. For each teacher-student pair, we select the value of $p_{fake}$ in $\{0.0, 0.1, 0.2, \ldots, 1.0\}$ that yields the highest validation set accuracy and report performance on the held-out test set. This results in the choice $p_{fake} = 0.8$ for the NIN-LeNet teacher-student pair and 0.2 for the other pairings.

Fig. 1a displays student test accuracy following each epoch of compression training with the $L^2$ logit-matching objective. In the end, both versions of GAN-MC significantly outperform compression on training data alone and training without compression ('Student Only'). The results are particularly striking for the mixture of GAN and training data which doubles the impact of training data compression. wIn this case, student accuracy increases by 10.5 percentage points (from 66.2% to 76.7%) with GAN-MC as opposed to 5.3 percentage points (from 66.2% to 71.5%) with training data alone. Table 1 reports comparable improvements for the NIN-LeNet teacher-student pairing when the cross-entropy compression objective is used. Indeed, the mixture of GAN and training data improves upon training data compression for all teacher-student pairings investigated.

At the start of the compression training in Fig. 1, compression with real data is more effective, presumably because the real training data provide a more faithful reflection of the real data distribution,

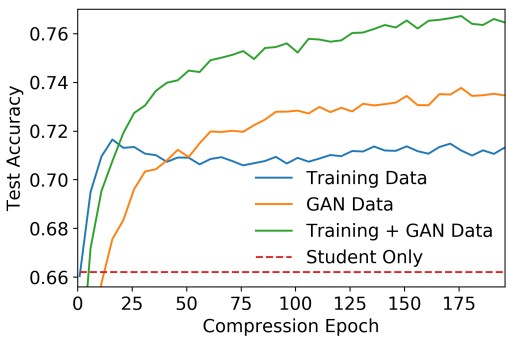 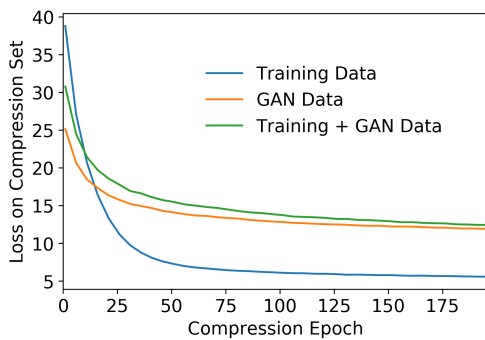

(a) Student classification accuracy on test set.

(b) Student $L^2$ training loss on compression set

Figure 1: Student performance for $L^2$ logit-matching compression as training progresses on CIFAR-10 (see Sec. 3). The teacher and student are NIN and LeNet with test accuracies of $78.1\%$ and $66.2\%$ (red dashed curve in (a)) when trained without compression. Compression is performed using only real training data (blue curve), only synthetic GAN data (green curve), or a mixture of training data and GAN data (orange curve, $p_{fake} = 0.8$). Results are averaged over 3 independent student training runs.

Table 1: CIFAR-10 image classification test accuracies for cross-entropy compression with various teacher and student neural net architectures. GAN-MC outperforms compression on training data alone and training without compression ('Student Only') in all cases. See Sec. 3 for more details.

|   | Teacher | Student | Teacher Only | Student Only | Student after Compression with Training Data | Training & GAN |
|---|---------|---------|---------|---------|---------|---------|
| 1 | NIN | LeNet | 78.1% | 66.2% | 71.0% | **75.3%** |
| 2 | ResNet-18 | 5-layer CNN | 94.2% | 78.8% | 84.4% | **86.6%** |
| 3 | WideResNet-28-10 | ResNet-18 | 95.8% | 94.2% | 94.3% | **95.0%** |

and the overfitting effect is not yet severe. Correspondingly, a quicker increase in test accuracy is observed at the start, as shown in Fig. 1a. After approximately 10 epochs, the influence of overfitting gradually increases and becomes dominant over the advantage of fidelity to the real data distribution. The compression set loss for real training data becomes significantly smaller than the loss with either version of GAN-MC, as shown in Fig. 1b, and the test accuracy stops increasing, as confirmed by Fig. 1a. Moreover, the teachers in our experiments yield $100\%$ accuracy on the training set but significantly lower accuracy on test datapoints, indicating a significant difference between the distributions of training and test set logit values and a disadvantage to relying wholly on training points. This dynamic illustrates the trade-off between GAN faithfulness to the real data distribution and the influence of overfitting and suggests that GAN-MC improves accuracy by mitigating overfitting to the compression set using a plentiful source of fresh and realistic (albeit imperfect) data.

**Effect of the GAN training proportion parameter $p_{fake}$.** Adopting the experimental setup of Fig. 1, we next examine how $p_{fake}$, the probability of selecting a GAN datapoint over a real datapoint when training the student, affects compression performance. We plot the dependence of trained student test accuracy on $p_{fake}$ in Fig. 2a. When $p_{fake} = 0$, only training data is used for compression; when $p_{fake} = 1$, only GAN data is used. Notably, every non-zero setting of $p_{fake}$ leads to improved accuracy over compression with the real training data alone, underscoring the value of GAN augmentation. Beyond this, we observe a non-monotonic but unimodal dependence on $p_{fake}$ with a combination of GAN and real datapoints providing significantly higher accuracy than GAN or real datapoints alone. This is consistent with a trade-off between the overfitting caused by training data reuse and the inability of a GAN to perfectly approximate the true data distribution.

**GAN-MC complements standard augmentation.** Our next experiment explores the impact of standard image augmentation on compression with and without GAN-MC. We follow the experimental setup of Fig. 1 but, during teacher and student training, introduce random image augmenta-

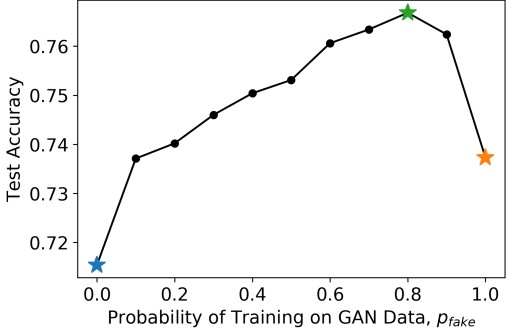

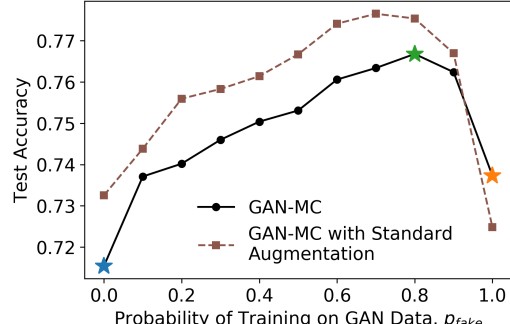

(a) **Effect of $p_{fake}$:** GAN-MC student test accuracy vs. probability $p_{fake}$ of training on GAN data. Compression using only training (resp. only GAN) data corresponds to $p_{fake} = 0$ (resp. $p_{fake} = 1$). The colored stars are the final values of the curves with the same color in Fig. 1a.

(b) **GAN-MC complements standard augmentation:** GAN-MC student test accuracy when teacher and student are trained with and without standard image augmentation (left-right flipping).

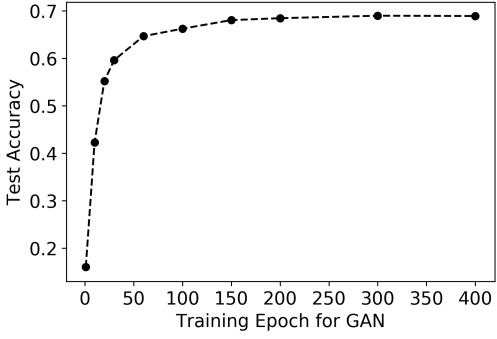

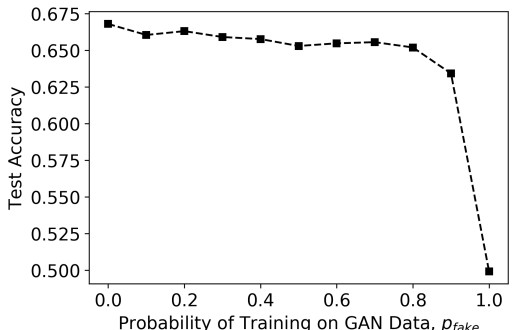

(c) **Quality matters:** GAN-MC student test accuracy ($p_{fake} = 1$) as a function of GAN quality, measured in GAN training epochs.

(d) **GAN-assisted supervised learning:** Student test accuracy when student is trained directly for the supervised learning task (without compression) and student training data is augmented with GAN data.

Figure 2: Student test accuracies in four experiments using the CIFAR-10 compression setup of Fig. 1. We report the average of 3 independent runs. See Sec. 3 for more details.

tions in the form of left-right image flips. In Fig. 2b, we see a clear benefit from introducing standard augmentation, and the greatest gain is realized when GAN and standard augmentation are combined.

**Quality matters.** To investigate the degree to which the quality of synthetic data affects compression improvement, we repeat the compression experiment of Fig. 1 using GAN data of varying quality and $p_{fake} = 1$. We use the number of training epochs for the GAN as a proxy for the GAN's quality. As shown in Fig. 2c, student test accuracy is greatly impaired by using a low-quality GAN trained for too few epochs. Fortunately, student test accuracy monotonically improves as the number of epochs and GAN fidelity increase.

**GAN-MC vs. GAN-assisted supervised learning.** In Sec. 2.3, we discussed the significant differences between GAN-MC and using GAN data to augment the training set for the original supervised learning problem. Fig. 2d shows that the same mixtures of GAN and training data that improve student compression performance in Fig. 2a actually impair accuracy when the student is trained without compression for the original supervised learning task.

## 4 RANDOM FOREST GAN-MC

We next use three tabular datasets from Kaggle and the UCI Machine Learning Repository to explore how GAN-MC performs when used to compress large random forests for binary classification. A

description of each dataset is given in Table 2. Higgs and MAGIC (MAGIC Gamma Telescope) are physics datasets from the UCI Machine Learning Repository (Dheeru & Karra Taniskidou, 2017). The nearly-class-balanced Higgs dataset was developed to learn whether a given observation was produced by a Higgs boson (Baldi et al., 2014). For our experiment, a subset of 200,000 class-balanced datapoints were selected uniformly at random. The target of the MAGIC dataset is the registration of high energy gamma particles in a gamma telescope. The Evergreen (StumbleUpon Evergreen) dataset is from Kaggle (https://www.kaggle.com/c/stumbleupon) and its target is whether webpages are evergreen or not. Following the feature extraction protocol in (Liu et al., 2017), we extract 29 continuous features for our regression. We split each datasets into training and test sets uniformly at random, with training split sizes given in Table 3.

In our experiments, the teacher is a random forest classifier with 500 trees, and the student is a regression random forest with one to 20 trees; both are trained using scikit-learn (Pedregosa et al., 2011) with all features considered for the best split (max_features = None) and default values for all other hyperparameters. The trees trained by the teacher and students have similar depth after training. When the student is trained without compression ('Student Only'), the class labels (0 and 1) are treated as real value targets. For the AC-GAN implementation in Keras, both the generator and the discriminator are one layer fully-connected neural network with 50 neurons and ReLU activation. We employed noise vectors $w \in \mathbb{R}^{100}$ and an Adam optimizer with learning rate 0.0002 and momentum term $\beta_1 = 0.5$.

We study three scenarios: compression using training data only, GAN data only or a mixture of training and GAN data. We generate $n_{fake} = 9\,n_{real}$ GAN datapoints for the compression set, where $n_{real}$ is the number of real training datapoints. The mixture compression set is generated by pooling the $n_{real}$ training datapoints and the $n_{fake}$ GAN datapoints together.

The results of compressing a random forest with 500 trees into a single decision tree are given in Table 3. We use test accuracy as our performance metric for the balanced Higgs dataset and test AUC for the unbalanced MAGIC and Evergreen datasets. We experiment with a variety of training dataset sizes, ranging from $n = 1k$ to $n = 100k$ to demonstrate the versatility of GAN-MC. For all datasets, compression with GAN data outperforms compression with training data and substantially outperforms the student model trained without compression. Moreover, for the Higgs dataset, the accuracy boost from GAN compression (62.1% to 68.5% on Higgs 100k) is 10 times the accuracy boost achieved using training data compression (62.1% to 62.7%).

The example of the Evergreen dataset is also enlightening. Compression with training data increases student test AUC from 0.731 to 0.856, and compression with only GAN data yields a further improvement to 0.882, nearly matching the 0.889 test AUC of the teacher. Remarkably, this is achieved with a single decision tree which demands 336 times less computation and storage space than the teacher at prediction time. The figure 336 comes from an assessment of student test-time speed-ups summarized in Fig. 3f. For each dataset, we identified the highest accuracy and most compressed students trained with and without GAN-MC and measured throughput as the time needed to compute predictions for $30,000$ test examples using one core of an Intel Xeon 6152 processor. At a cutoff of $1.2\%$ excess test error, we observe speed-ups ranging from 25 to 336-fold.

Figs. 3a-3d displays student test performance as a function of the number of trees in the student forest. For each dataset save Higgs 1k, compressing with GAN data offers the best (or nearly the best) performance for all forest sizes. Indeed, for the Evergreen and MAGIC datasets, near-maximal performance is achieved by a single GAN-MC decision tree, with additional trees yielding relatively minor performance gains. For Higgs 1k, the combination of training and GAN data offers the best performance for all multi-tree forests, with an accuracy boost consistently 2-4 times that of compression with training data alone.

Table 2: Description of tabular datasets used for random forest GAN-MC.

| Dataset | # Datapoints | # Features | Class Imbalance |
|---|---|---|---|
| Higgs | 200k | 28 | 0: 50.0%; 1: 50.0% |
| MAGIC | 19k | 11 | 0: 35.2%; 1: 64.8% |
| Evergreen | 7k | 29 | 0: 48.7%; 1: 51.3% |

Table 3: Test accuracy (Higgs) and test AUC (Evergreen and MAGIC) of the learned student in random forest compression. Here a forest with 500 trees is compressed into a single decision tree.

| Dataset | Training Data Size | Teacher Only | Student Only | Student after Compression with | | |
|---|---|---|---|---|---|---|
| | | | | Training Data | GAN Data | Training & GAN |
| Higgs | 1k | 66.4% | 56.2% | 56.5% | **59.0%** | 57.7% |
| | 100k | 72.6% | 62.1% | 62.7% | **69.6%** | 64.7% |
| MAGIC | 10k | 0.935 | 0.785 | 0.895 | **0.918** | 0.912 |
| Evergreen | 5k | 0.889 | 0.731 | 0.856 | **0.882** | 0.849 |

**GAN-MC vs. GAN-assisted supervised learning.** Consistent with our discussion in Sec. 2.3 and our findings in Fig. 2d, Fig. 3e shows that the same GAN data that substantially improves student compression performance in Fig. 3d harms or scarcely improves test AUC when the student is trained without compression for the original MAGIC supervised learning task.

## 5 EVALUATING GANS WITH A COMPRESSION SCORE

The evaluation of synthetic datasets is an important but challenging task. Two criteria commonly considered essential for a high-quality synthetic dataset are datapoint diversity and discriminability. The most widely used GAN quality measure, the Inception Score of Salimans et al. (2016), measures across-class diversity but does not account for within class diversity. In addition, the Inception Score measures a form of discriminability based on the predictions of a pre-trained neural network but is easily misled by datapoints that elicit high confidence predictions without resembling real data. For example, if the classification loss $L_{class}$ is heavily upweighted relative to the source loss $L_{source}$ while training an AC-GAN, the generator will be more likely to produce feature vectors classified with high confidence by neural networks. As we will demonstrate in Sec. 5.2, such feature vectors need not resemble real data but will nevertheless receive high Inception Scores (which should be reserved for high-quality datasets). To account for both discriminability and diversity in a more robust and holistic manner, we propose to use the performance of a student trained on GAN data as a measure of GAN dataset quality.

### 5.1 THE COMPRESSION SCORE

To evaluate the quality of a generated dataset $\mathcal{D}$ relative to a real dataset $\mathcal{D}_{real}$, we define a *Compression Score* based on the test accuracy $acc(\mathcal{D})$ of a student trained with compression set $\mathcal{D}$ to mimic a teacher pre-trained on $\mathcal{D}_{real}$:

$$\texttt{CompressionScore}(\mathcal{D}; \mathcal{D}_{real}) = \frac{acc(\mathcal{D}) - acc_{mode}}{acc(\mathcal{D}_{real}) - acc_{mode}}.$$

The term $acc_{mode}$ represents the accuracy obtained by always predicting the most common class in $\mathcal{D}_{real}$. A higher Compression Score is designed to indicate a higher quality dataset $\mathcal{D}$.

The Compression Score declares a synthetic dataset to be of higher quality if a compressed model trained only on that data achieves higher accuracy on real test data. Increased within-class diversity, increased across-class diversity, and increased discriminability all tend to increase the Compression Score, as they enable the student to more accurately mimic the teacher's output across all classes. However, crucially, the Compression Score is only impacted by aspects of discriminability and diversity that matter for performance on real test data. Hence, unlike the Inception Score which is completely determined by the idiosyncratic output of an imperfect network, the Compression Score is robust to the idiosyncratic preferences of an imperfect teacher or student. In particular, we would not expect a student trained on unrealistic or adversarial synthetic data to perform well on real test data even if it very accurately mimics the teacher's predictions on such data.

By design, the Compression Score equals 1 for the real dataset $\mathcal{D}_{real}$ and tends to 0 as the synthetic data distribution diverges from the real data distribution one. As discussed in Sec. 3 and highlighted in Fig. 1, we have found empirically that, in the initial epochs of compression training, test accuracy typically increases more rapidly when training data is used than when synthetic data is used. As a

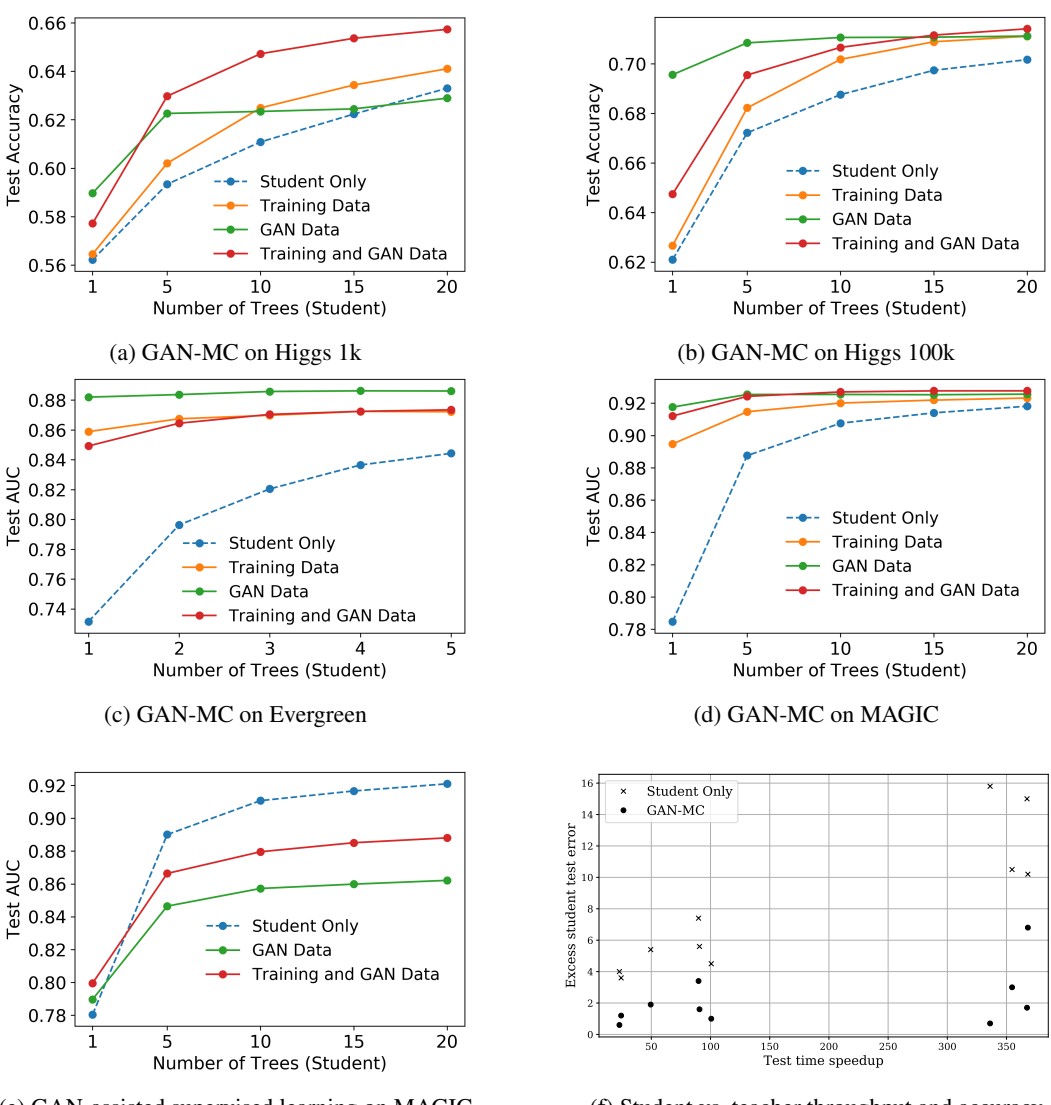

Figure 3: (a-e) Test accuracy (Higgs) and test AUC (Evergreen and MAGIC) of the learned student in random forest compression. Here a 500 tree forest is compressed into a compact random forest. When the student is trained directly on training data without compression ('Student Only'), its performance is given by the blue dashed curve. (a-d) Compression is carried out on only training data (orange curve), on only GAN data (green curve) or on a mixture of the training data and GAN data (red curve). (e) Without compression, the student is trained for the original supervised learning task using only GAN data (green curve) or a mixture of the training data and GAN data (red curve). (f) For all datasets, GAN-MC students increase test time throughput (i.e., number of test examples processed per second) 25 to 336-fold over teacher with less than $1.2\%$ loss of accuracy.

result, a student restricted to one epoch of training tends to produce Compression Scores in $[0, 1]$. To exploit this desirable property and simultaneously reduce Compression Score evaluation time, we train each student for only one epoch in our experiments.

## 5.2 EVALUATING GANS: AN ILLUSTRATION WITH CIFAR-10

To illustrate the potential benefit of the Compression Score over the commonly-used Inception Score, we reinstate the CIFAR-10 experimental setup of Fig. 1. The standard error is obtained from 3 independent runs. We evaluate the compression score on real data, well-trained GAN data

Table 4: Inception and Compression Scores for CIFAR-10 images; larger scores should signify higher quality images. Inferior data generated by training well-trained GAN for 10 additional epochs using only the classification objective $L_{class}$ (see Sec. 5.2). Inception Score increases for inferior images despite evident unrealistic artifacts. Compression Score decreases for inferior images.

| Real Data | Well-trained GAN | Inferior GAN |
|:---:|:---:|:---:|
|  |  |  |
| Inception: $11.2 \pm 0.1$ | Inception: $5.80 \pm 0.06$ | Inception: $5.93 \pm 0.06$ |
| Compression: $0.994 \pm 0.003$ | Compression: $0.778 \pm 0.002$ | Compression: $0.702 \pm 0.002$ |

(i.e., data from the AC-GAN described in Sec. 3), and inferior data which have high confidence classifications under the teacher network but do not resemble real data. The inferior data are generated by training the well-trained AC-GAN for 10 additional epochs using only the classification objective $L_{class}$ given in Eq. 2a. That is, both the generator $G$ and discriminator $D$ are trained to maximize $L_{class}$, while ignoring the traditional GAN objective component $L_{source}$.

In Table 4, the quality of the GAN data degrades noticeably after the additional training with only $L_{class}$. Unrealistic artifacts are evident in the inferior images, but the Inception Scores of those images are higher than those of the well-trained images. In contrast, the Compression Score decreases in accordance with our expectations as the GAN images become evidently worse. To highlight the practicality of the Compression Score, we also performed a timing comparison of Inception and Compression Score evaluations on this dataset. Using the Inception Score code of (Salimans et al., 2016) and an NVIDIA Tesla V100 GPU, the Inception Score required 1436.6s and the Compression Score 350.1s.

## 6 RELATED AND FUTURE WORK

To reduce the deployment costs of expensive machine learning classifiers, we introduced GAN-assisted model compression (GAN-MC) as a straightforward way to improve teacher-student compression. We demonstrated the benefits of GAN-MC for both image and tabular data classifiers and developed a new Compression Score for evaluating the quality of synthetic datasets. While we have focused on improving the popular teacher-student paradigm of model compression, we would be remiss to not mention alternative, model-specific approaches to reducing deployment costs, including parameter sharing (Chen et al., 2015), network pruning (Han et al., 2015), and network parameter prediction (Denil et al., 2013) for DNNs and indicator function selection (Joly et al., 2012), pre-pruning (Begon et al., 2017), and probabilistic modeling and clustering (Painsky & Rosset, 2016; 2018) for random forests.

A number of exciting opportunities for future work remain. For example, GAN-MC is readily integrated into more complex approaches to teacher-student compression that currently reuse the teacher's training data for compression. Prime examples are the recent approaches of Wang et al. (2018) and Xu et al. (2018) which train an assistant network to discriminate between teacher and student outputs. In addition, GAN development for tabular data has received much less attention than GAN development for image data, and we anticipate that significant improvements over the AC-GANs used in our experiments will result in significant performance benefits for GAN-MC.

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
