# OpenReview forum: "Model Compression with Generative Adversarial Networks"
_ICLR.cc/2019/Conference_

### Official Review · AnonReviewer2 · 2018-11-01
**interesting idea, some important experiments missing**

**Rating:** 5
**Confidence:** 4

**Review:**

I like this paper. What the authors have done is of high quality. It is well written and clear. However, quite a lot of experiments are necessary to make this paper publishable in my opinion.

Strenghts:
- The idea to use a GAN for model compression is something that many must have considered. It is good to see that someone has actually tried it and it works well.
- I think the compression score is definitely an interesting idea on how to compare GANs that can be of practical use in the future.
- The experimental results, which are currently in the paper, largely support what the authors are saying.

Weaknesses:
- The authors don't compare how good this technique is in comparison to simple data augmentation. My suspicion is that the difference will be small. I realise, however, that the advantage of this method over data augmentation is that it is harder to do it for tabular data, for which the proposed method works well. Having said that, models for tabular data are usually quite simple in comparison to convnets, so compressing them would have less impact.
- The experiments on image data are done with CIFAR-10, which as of 2018 is kind of a toy data set. Moreover, I think the authors should try to push both the baselines and their technique much harder with hyperparameter tuning to understand what is the real benefit of what they are proposing. I suspect there is a lot of slack there. For comparison, Urban et al. [1] trained a two-layer fully connected network to 74% accuracy on CIFAR-10 using model compression.

[1] Urban et al. Do Deep Convolutional Nets Really Need to be Deep (Or Even Convolutional)? 2016.

---

> ### Author Response · Authors · 2018-11-27
> **New experiments and tabular data impact**
>
> We thank the reviewer for the positive and constructive feedback; we provide detailed responses below and have updated our manuscript accordingly.
>
> -Simple data augmentation
> A primary contribution of this work is developing an improved compression approach that works for both tabular and image data alike. We have added an experiment showing that standard image augmentation complements GAN augmentation: while standard augmentation improves compression on training data alone, the greatest gain is achieved by applying standard augmentation on top of GAN-MC. For tabular data, the reviewer is correct that it is unclear how to define an appropriate analogue of standard image augmentation, but GAN-MC is equally applicable.
>
> -Tabular data impact
> In the revision, we highlight the literature attesting to the importance of compressing tabular random forests for memory and computation-constrained environments (e.g., Model Compression, Compressing Random Forests, Globally Induced Forest: A Prepruning Compression Scheme, and L1-based compression of random forest models). GAN-MC is especially effective in this setting, where we achieve up to 375-fold reductions in test time computation and storage by compressing a 500-tree forest into a single decision tree of similar quality.  To clarify the impact on test-time throughput, we now include displays of teacher vs. student test time throughput in Fig. 3f.
>
> -Additional experiments
> In response to all reviewers’ suggestions, we now explore how student performance varies as a function of GAN quality in Fig. 2c and highlight the difference between using GAN-augmentation for compression and for the original supervised learning task in Fig. 2d for neural networks and in Fig. 3f for random forests.

---

### Official Review · AnonReviewer1 · 2018-11-01

**Rating:** 6
**Confidence:** 4

**Review:**

Summary:
The paper proposes an approach for improving standard techniques for model compression, i.e. compressing a big model (teacher) in a smaller and more computationally efficient one (student), using data generated by a conditional GAN (cGAN). The paper suggests that the standard practice of training the student to imitate the behavior of the teacher *on the same training data* that the teacher was trained on is problematic and can lead to overfitting. Instead, the paper proposes learning a conditional GAN, which can potentially generate large amounts of realistic synthetic data, and use this data (in addition to original training data) for model compression.
Experimental results show that this idea seems to improve the performance of convnet student models on CIFAR-10 classification and random forest student models on tabular data from UCI and Kaggle.
Another contribution of the paper is to propose an evaluation metric for generative model, called the compression score. This score evaluates the quality of generated data by using it in model compression: “good” synthetic data results in a smaller gap in performance between student and teacher models.

Strengths:
-	The paper sheds a light on an interesting aspect in model compression. The idea of teaching a student model to imitate behavior of the teacher model on *new* data is interesting. In fact, it emphasizes the fact that we are mostly interested in imitating the teacher model’s capability of generalizing to new examples rather than overfitting to training examples.
-	Experiments show that for several settings (model class, architecture and datasets), using synthetic data by a cGAN can be useful in reducing the gap between student and teacher models.
-	The paper is clearly written and easy to follow.

Weaknesses:
-	The claim that reusing the same training data used for training the teacher model in model compression can lead to overfitting of student model is not very obvious and needs more experimental evidence in my opinion. One way to test this is to use some unseen real data (e.g. validation or a held-out part of training data) for model compression, and showing that it can indeed help in improving student performance.
-	The claim that cGAN can generate “infinite” amount of realistic data is too strong. In light of some well-known problems of GANs such as mode collapse [2] and low-support learned distributions [1], this assumption seems unrealistic. In fact, it is not too obvious how synthetic data by a generative model learned on *same training data as the teacher* can provide any additional information to real data.
-	While the idea of the proposed evaluation metric seems interesting, I believe it is not very practical, because:
1.	It is computationally intensive (requires training a model from scratch on fake data)
2.	It relies on performance of the compression mechanism, which might also have some idiosyncrasies that prefer some features in synthetic data which do not necessarily correspond to quality of generated data.

Questions/Suggestions:
-	In addition to using held-out real data for model compression as suggested above, a useful baseline could be using standard data-augmentation techniques in model compression.
-	What would happen if a student model is very small and cannot possibly overfit training data? Would using synthetic data be still useful there?
-	I am actually confused about a claim made when presenting compression score in Section 5. The paper claims that the best compression score is 1 (training student model on real data), while the paper shows that in fact, good synthetic data should produce *better* accuracy than using real data. I would appreciate if authors can clarify this point.

Overall recommendation:
While the paper presents an interesting problem in model compression, I’m leaning towards rejecting the paper because of the weaknesses mentioned above. That being said, I am happy to reconsider my decision if there is any misunderstanding on my part.

References:
[1] Arora, Sanjeev, and Yi Zhang. "Do GANs actually learn the distribution? an empirical study." arXiv preprint arXiv:1706.08224 (2017).
[2] Goodfellow, Ian. "NIPS 2016 tutorial: Generative adversarial networks." arXiv preprint arXiv:1701.00160 (2016).


-----

Updated score and posted a comment to author response.

---

> ### Author Response · Authors · 2018-11-27
> **Clarifications and new experiments**
>
> -Overfitting and weak students
> Even when the compression set is large, overfitting can occur because the distribution of teacher logits looks very different for its training points than for fresh test points due to the teacher’s overfitting to the training set. If the student is strong enough to distinguish between the train and test logit distributions, then we expect augmentation to help. If the student is so weak that it cannot distinguish between the train and test distributions and the training set is large, then augmentation may not help. We see an example of this in the error curves of Fig. 1. In the first few epochs of student training, the real training data leads to a greater decrease in test loss than the GAN augmented data, but the GAN augmented data eventually leads to a better test loss. Here, the network constrained to train for only one epoch is an example of a weak student with less discriminating ability than the fully trained network.
>
> -Suboptimal compression with training data
> Augmenting the compression set with held-out data from the training distribution has been shown to improve compression in prior work (see Fig. 4 in Bucila et al. and Sec. 4 of Ba and Caruana). We view this as the ideal case. All of our experiments were designed to provide even stronger evidence that compression with training data alone is suboptimal: not only can compression be improved, it can be improved in the common scenario where one does not have access to fresh real data.
>
> -Infinite data
> This is a very good point. We have replaced “infinite” with “auxiliary” throughout to avoid any confusion and now emphasize that our approach does not hinge on having an infinite amount of data but rather benefits from access to an auxiliary source of realistic data.
>
> -Value of synthetic data
> We agree that the idea of using GAN datapoints to improve compression is counterintuitive, and we now clarify in the revision why it is sensible: The goal in model compression is to approximate the teacher prediction function g which maps from inputs to predictions z. Because the teacher is a function of the training data alone, g itself is a functional of the training data alone and is otherwise independent of the unknown distribution that generated that data. In addition, because we have access to the teacher, we have the freedom to query the function g at any point, and hence our information concerning g is limited only by number of queries we can afford. When we generate a new query point x, we can observe the actual target value of interest, the teacher’s prediction g(x) (this is not true for the original supervised learning task, where no new labels can be observed). We believe these properties make the model compression task more tractable than the original supervised learning task and ideal for data augmentation with generative models. Similar rationale is given in Bucila et al.
>
> ln addition, we believe that the best proof that GAN-MC is valuable is to demonstrate that it works in practice. In our submission we show that, in multiple experiments on multiple datasets with a variety of students and teachers, true test set performance improves when GAN-MC is used. Especially compelling is Figure 2a which shows that no matter what non-zero value of pfake we use, GAN-MC improves upon compression with training data alone.
>
> -Compression Score practicality
> The compression score is quite practical to compute as only one epoch of training is conducted; in our Sec. 5 experiment, the Inception Score evaluation required 1436.6s using the code of Salimans et al., while the Compression Score required 350.1s. Both evaluations were done in Tensorflow using an NVIDIA Tesla V100 GPU.
>
> -Idiosyncrasies
> We agree that any imperfect prediction rule is subject to idiosyncrasies. However, as we clarify in the revision, the Inception Score is completely determined by the idiosyncratic output of the network, while the Compression Score is determined by performance on real test data, so the Compression Score will only increase if the synthetic data leads to better classification of real data.
>
> -Standard data augmentation
> We have added an experiment showing that standard image augmentation complements GAN augmentation: while standard augmentation improves compression on training data alone, the greatest gain is achieved by applying standard augmentation on top of GAN-MC. For tabular data, as Rev. 2 notes, it is unclear how to define an appropriate analogue of standard image augmentation, but GAN-MC is equally applicable.
>
> -Compression score range
> We apologize for the confusion and clarify in the revision. The Compression Score can take on values larger than 1; however, we have found empirically (see, e.g., Fig. 1) that accuracy after the first epoch is typically superior when training data is used than when GAN data is used. This was part of our rationale for using only 1 epoch of training to define the Compression Score.

---

> > ### Comment · AnonReviewer1 · 2018-11-28
> > **Thanks for the detailed response**
> >
> > Additional comments:
> >
> > - Thanks for pointing our these results in previous work of Bucila et al. I think this deserves to be explicitly mentioned in your paper, because it provides direct evidence for your claim.
> >
> > - I think you should report the effect of data-augmentation *alone*. Also, adding more than one data-augmentation strategy would strengthen the result (especially for image data, where there are plenty of effective methods). That being said, I agree with a point raised reviewer Reviewer-2 that for other kinds of data (e.g. tabular data) there might not be effective ways for data-augmentation.
> >
> > - I am not sure how you compare the execution time of inception score compared to your methods. Fundamentally, inception score requires only a forward-pass in a pretrained model, which can be done with a small number of examples (e.g. 1K or 5K). Training a model from scratch would require a forward and backward pass, and probably on much more data, but I guess the model is much smaller than inception. Also, I'm not sure it's clear from the paper that you do only one epoch of training.
> >
> > I am going to raise my score to 6, because I think the paper has some interesting aspects to it. I still believe it's a borderline paper, especially that I'm not convinced of the effectiveness of compression score and that GANs can be substantially more effective than data-augmentation.

---

> > > ### Author Response · Authors · 2018-11-29
> > > **Additional clarifications**
> > >
> > > Thank you for your responses; we include a few additional clarifications below.
> > >
> > > -Value of augmentation with test data
> > >
> > > Thanks for this suggestion; in the final version we will update our references to these works to clarify that they provide direct evidence for the value of augmenting the compression set with test data.
> > >
> > > -Standard augmentation
> > >
> > > We believe our initial response was unclear.  In the revision, we report the effect of data augmentation alone (as well as data augmentation on top of GAN-MC) in Fig. 2b.  Standard data augmentation corresponds to the point in the GAN + standard augmentation curve with p_fake = 0.  We see that standard data augmentation alone increases student test accuracy to 73.3% (vs. 76.7% for GAN-MC alone).  In addition, any augmentation that can be performed on the training data can also be applied to GAN data for added diversity; in this case standard augmentation on top of GAN-MC has a maximum student test accuracy of 77.7%.
> > >
> > > We agree that it will be valuable to evaluate additional image augmentation schemes as well.  In the final version, we will report the performance of those schemes both alone and applied on top of GAN-MC.
> > >
> > > -Inception vs. Compression
> > >
> > > To compute the Inception score we perform one forward pass on the synthetic 50K images.  To compute the Compression Score we compute one forward and one backward pass on the same 50K images to train the student for a single epoch.  We finally perform one forward pass on 10K test images to compute student test accuracy.  The reviewer is correct that our use of a much smaller student network (the NIN net from Sec. 3) also shortens the computation time.  We will clarify these points in the final version.
> > >
> > > In the revision, we state that only one epoch of training is performed in Sec. 5.1, but in the final version we will highlight this in the following subsection as well so that there is no misunderstanding.

---

### Official Review · AnonReviewer3 · 2018-11-03
**More explanation and clarification on experiments required**

**Rating:** 5
**Confidence:** 4

**Review:**

This paper focused on training a small network with a pre-trained large network in a student-teacher strategy, which also known as knowledge distillation. The authors proposed to use a separately trained GAN network to generate synthetic data for the student-teacher training.

The proposed method is rather straightforward. The experimental results look good, GAN generated data help train a better performed student in knowledge distillation. However, I have concerns about both motivations and experiments.

1. The benefits of GAN for generating synthetic data to assist supervised training are still mysterious, especially when GAN is separately trained on the same dataset without more information introduced. I would love the authors to clarify why GAN generated data are particularly effective for knowledge distillation. Does GAN generated data also help standard supervised training? I would expect following experiments: use mixture of training and GAN data to train teacher and student network by standard supervised loss without knowledge distillation, and compare with values in table 1.

2. The performance of the proposed method depends on the quality of GAN. To help me further understand the quality of GAN, I hope to see the following experiments to compare with scores in table 1.
i) The accuracy of supervised trained teach and student on GAN generated image.
ii) The classification accuracy on test data by the classifier trained in AC-GAN.

3. I would like the authors to clarify their experiments to convince me the comparison in table 1 is fair.
i) How many data and iterations in total are used for standard training and knowledge distillation with/without GAN data? Does the better performance come from synthetic data, or come from exploiting more data and training for longer time?
ii) Related to i), In figure 1 (a), how many data and iteration for each epoch? It would help if the standard supervised training curve for student can be provided.
iii) The experiments have a lot of hyperparameters, for example, the weight \alpha, the temperature T,  optimizer, the learning rate, learning rate decay, the probability p_fake. These hyperparameters are different for each experimental setting. How are they chosen?

4. Please explain conceptually why the proposed compression score is better than inception score.

5. The paper is missing a conclusion section. The following papers introduce adversarial training for knowledge distillation. Though it is not necessary to compare with them in experiments as they are complicated method and the usage of GAN is different from this paper, I think it is still worth to mention them in related work.
Wang et al. 2018 Adversarial Learning of Portable Student Networks
Xu et al. 2018 Training Student Networks for Acceleration with Conditional Adversarial Networks

================ after rebuttal ====================
I appreciate the authors' response and slightly raise the score. It is a good rebuttal and it has clarified several things. I like the authors' explanation on why GAN is particularly good in a student-teacher setting. The explanation reminds me of the mixup data augmentation paper from last year. I also like the additional experiments which clearly show the benefits of GAN data augmentation.

However, I still think it is borderline for several reasons.
1. As the other reviewer has pointed out, CIFAR-10 is a bit too toy and some models (like LeNet for Figure 2) cannot really show the advantage of the method. I would suggest try ImageNet, and use more recent networks for ablation study.
2. As the other reviewer has pointed out, the compression ratio can be impractical. The compression ratio  depends on student-teacher training, which can take a relatively long time.
3. I would suggest the following experiments that may strengthen the paper. I would consider these as a plus, not necessarily related to my current evaluation.
i) Try not use GAN, but use mixup (linear interpolation of samples) as data augmentation, and go through the student-teacher training.
ii) Try evaluate the effect of generator structure for data augmentation. Does the generator have to be very strong? The GAN generated results did not improve supervised learning may suggest the generator is not necessarily to be strong.

---

> ### Author Response · Authors · 2018-11-27
> **Experiment explanations and clarifications**
>
> We thank the reviewer for the positive and constructive feedback; we provide detailed responses below and have updated our manuscript accordingly.
>
> -Why GAN generated data are particularly effective for knowledge distillation / Accuracy of GAN-augmented supervised learning
>
> We clarify in the revision that there is an important difference between using GANs for model compression and using GANs for the original supervised learning task.  The goal of the original supervised learning task is to approximate the ideal mapping f* between inputs x and outputs y.  This ideal f* is a functional of the true but unknown distribution underlying our data, and our information concerning f* is limited by the real data we have collected.  The goal in model compression is to approximate the teacher prediction function g which maps from inputs to predictions z.  Because the teacher is a function of the training data alone, g itself is a functional of the training data alone and is otherwise independent of the unknown distribution that generated that data.  In addition, because we have access to the teacher, we have the freedom to query the function g at any point, and hence our information concerning g is limited only by number of queries we can afford.  In particular, when we generate a new query point x, we can observe the actual target value of interest, the teacher’s prediction g(x); this is not true however for the supervised learning task, where no new labels can be observed.  We believe these properties make the model compression task a more tractable one and one that is ideal for data augmentation with generative models.
>
> Following the reviewer’s suggestion, we complement this explanation with two new experiments (one for DNNs in Fig. 2d and one for random forests in Fig. 3e) showing that the same GAN data that greatly improves distillation performance either harms or scarcely improves performance in the supervised learning task.
>
> -GAN quality matters
> The reviewer is correct that the performance of GAN-MC depends on the quality of the GAN; to make this clearer we have introduced a new “quality matters” experiment demonstrating how student performance varies as a function of the number of GAN training iterations.
>
> -Fairness of Table 1 comparison
> We address all of these questions in the revision.  For image data, we use the same batch sizes and process the same number of batches for distillation with and without GAN data.  The improved performance apparently comes from the exposure to synthetic data in addition to the available training data.  Interestingly, in Fig. 2a we see improved performance for every non-zero value of p_fake.  For tabular data, we explicitly augment the training dataset of size n_train with 9n_train synthetic datapoints and run the default Random Forest scikit-learn training code.  All hyperparameters except p_fake were set to the default values recommended in (Li, 2018).  We select p_fake from the values {0, 0.1, 0.2, …, 1.0} using a validation set and report performance on the test set.
>
> -Compression vs. Inception
> We clarify the conceptual advantages of the compression score over the inception score in the revision.  The Inception Score measures across-class diversity but does not account for within class diversity.  In addition, the Inception Score measures a form of discriminability based on the predictions of a pre-trained neural network but is easily misled by datapoints that elicit high confidence predictions without resembling real data.  Because the Compression Score is determined by student performance on real test data, it benefits from both within and across-class diversity (this typically leads to a higher-quality students) and prefers datapoints that improve real test set error (students trained on less realistic datapoints tend to yield worse test error).
>
> -Conclusion
> We have added a conclusion section and discuss the mentioned related work therein.

---

> ### Author Response · Authors · 2018-11-30
> **Additional clarifications**
>
> Thank you for your responses and your suggestions to further improve the paper; we include some additional clarifications below.
>
> -Practicality of compression score
>
> The compression score is quite practical to compute as only one epoch of training is conducted (this is stated and motivated in Sec. 5.1, but in the final version we will highlight this in the following subsection as well so that there is no misunderstanding).  In our Sec. 5 experiment in the revision, we report that the Inception Score evaluation required 1436.6s using the code of Salimans et al., while the Compression Score required 350.1s. Both evaluations were done in Tensorflow using an NVIDIA Tesla V100 GPU.
>
> To compute the Inception score we perform one forward pass on the synthetic 50K images.  To compute the Compression Score we compute one forward and one backward pass on the same 50K images to train the student for a single epoch.  We finally perform one forward pass on 10K test images to compute student test accuracy.  Our use of a much smaller student network (the NIN net from Sec. 3) also shortens the relative computation time.  We will clarify these points in the final version.
>
> -Standard data augmentation
>
> In the revision, we report the effect of data augmentation alone (as well as data augmentation on top of GAN-MC) in Fig. 2b.  Standard data augmentation corresponds to the point in the GAN + standard augmentation curve with p_fake = 0.  We see that standard data augmentation alone increases student test accuracy to 73.3% (vs. 76.7% for GAN-MC alone).  In addition, any augmentation that can be performed on the training data can also be applied to GAN data for added diversity; in this case standard augmentation on top of GAN-MC has a maximum student test accuracy of 77.7%.
>
> We agree that it will be valuable to evaluate additional data augmentation schemes as well, such as the mixup data augmentation you mentioned.  In the final version, we will report the performance of those schemes both alone and applied on top of GAN-MC.
>
> -GAN strength
>
> While we have not yet explored varying the GAN architecture, we have found that the quality of the GAN generator has a significant impact on student performance.  For example, in Fig 2c, student test accuracy ranges from .1 to .7 as a function of the quality of the AC-GAN used to generate the synthetic data.  However, notably, compression performance does not depend on the quality of the synthetic class labels associated with each GAN point (while supervised learning performance certainly does).

---

### Meta-Review · Area_Chair1 · 2018-12-13
**Paper could be strengthened by evaluations on large-scale tasks**

**Confidence:** 4
**Recommendation:** Reject

**Metareview:**

The authors propose a scheme to compress models using student-teacher distillation, where training data are augmented using examples generated from a conditional GAN.
The reviewers were generally in agreement that 1) that the experimental results generally support the claims made by the authors, and 2) that the paper is clearly written and easy to follow.
However, the reviewers also raised a number of concerns: 1) that the experiments were conducted on small-scale tasks, 2) the use of the compression score might be impractical since it would require retraining a compressed model, and is affected by the effectiveness of the compression algorithm which is an additional confounding factor. The authors in their rebuttal address 2) by noting that the student training was not too expensive, but I believe that this cost is task specific. Overall, I think 1) is a significant concern, and the AC agrees with the reviewers that an evaluation of the techniques on large-scale datasets would strengthen the paper.